# Catalytic Effect of Green Human Resource Practices on Sustainable Development Goals: Can Individual Values Moderate an Empirical Validation in a Developing Economy?

**Jiakun Liu [1,2], Xinxiang Gao [3], Yi Cao [4], Naveed Mushtaq [5,\*], Jiuming Chen [6] and Li Wan [2]**

1 Business School, University of International Business and Economics, Beijing 100029, China
2 School of Economics and Management, Shandong Youth University of Political Science, Jinan 250103, China
3 Graduate School of Management, SEGI University, Kota Damansara PJU 5, Petaling Jaya 47810, Selangor, Malaysia
4 School of Management, Hebei Finance University, Baoding 071051, China
5 Noon Business School, University of Sargodha, Sargodha 40100, Pakistan
6 China Xiongan Group, Human Resources Department, Baoding 071700, China
* Correspondence: naveed.mushtaq@uos.edu.pk

**Abstract:** Utilizing the framework of the theory of planned behavior, a new model has been extracted and validated empirically to explore the role of green human resource management (GHRM) practices in the attainment of the sustainable development goals (SDGs) among textile manufacturing firms. Therefore, this research study is the first attempt to empirically investigate the impact of green skills and employee green attitudes that may proffer a better explanation of the nature of the relationship between GHRM and the SDGs, proposing an inclusive re'search model on the effect of GHRM on the SDGs in the hi-tech manufacturing industry. Questionnaires were distributed to 465 textile firms; however, of those returned, only 197 surveys completed in all respects were used for further empirical investigation. PLS-SEM was used to analyze the data to assess the validity and reliability of the instrument. The outcomes of the study affirmed the theoretical model according to which GHRM has a positive association with employee green attitudes, employee skills, decent work, and sustainable consumption and production behavior. Employee green attitudes and decent work had a positive effect on sustainable consumption and production behavior. It is also beneficial to identify significant moderators to explain the processes and circumstances through which the attitudes of employees are transformed into the desired behaviors. Therefore, individual green values were taken as moderators in the study to assess how they impacted the relationship between GHRM and employee green attitudes, as well as that between employee green attitudes and sustainable consumption and production behavior. The results also revealed that an employee's green attitude acted as a partial mediator between GHRM and decent work. Moreover, employee green attitudes and employee skills fully mediated between GHRM and sustainable consumption and production behavior.

**Keywords:** green human resource; planned behavior; attitude; individual values; sustainable consumption; decent work

## 1. Introduction

The rapid growth in human economic expansion has had disastrous effects on natural resources and the environment [1]. The social efforts and actions to redress the devastating situation have been minimal. The ever-increasing quest to develop well-developed economies towards industrialization in southeast Asia has raised serious questions over the transition of these economies towards a greener, low-carbon, and circular economy [2]. This major factor has led many multinational organizations to shift their focus from financial returns to other environment- and welfare-related outcomes viz. commitment to

environmental and social outcomes, an increase in social responsibility, and sustainable performance [3–6]. Nowadays, the critical question deals with how organizations working in developing countries are achieving their economic goals and how they can participate in achieving sustainable development [7]. Environment, economy, and society are the three fundamental pillars of sustainable development [8–10]. The lack of integration in terms of policies, strategies, and their implementation across different sectors has been the primary cause of the failure of all those approaches previously used for achieving sustainable development [11]. The sustainable development goals are a network of targets that provide unique integration and policy coherence among different sectors. These goals can be seen as a network as they are uniquely connected.

The SDGs are based on the five Ps, i.e., "People, Planet, Prosperity, Peace and last but not the least Partnership" [12], and are the contributing factors for creating a greener environment [13]. As far as people and their prosperity is concerned, the SDGs aim to formulate strategies that can elevate the efficiency of resource allocation, sustainable development, combined welfare, and a decent working environment. For business, the mission of the SDGs for organizations is to ensure a sound and well-educated environment. It also enhances the awareness of employees of the need to increase their productivity and create proactive citizens that can make a positive contribution to society. The collaboration of these agents is crucial to attaining SCPB. The agenda is created by the individuals for themselves, and it is they who are expected to ensure its success [14]. Human resource development can be considered as one fundamental department/area to complete the goals at the organizational level.

Therefore, the human element plays a dual role, as it is the initiator as well as the recipient in the accomplishment of SDGs. According to the observations of various business disciplines related to manufacturing and operational activities, the human factor is the primary source of the interconnection between HRM and the SDGs, as the attitudes, behavior, and resource consumption of people have a direct influence on the ecological and social practices [15]. Moreover, research on the impact of human resource management on consistent and sustainable growth remains scarce [16]. Awareness about the environment starts with the "green movement", which involves social and ecological aspects [16]. Similarly, sustainable human resource management (SHRM) provides a clear example of the inclusion of such sustainable practices into business [17]. Moreover, striking an equilibrium between commercial growth, the protection of environmental resources, and the achievement of organizational targets is the primary objective of sustainable human resource management [18].

There is a means-to-an-end kind of association between SHRM and the SDGs [19,20]. These days, businesses are becoming more and more conscious of the significance of social, ethical, and environmental goals. Organizations are adopting new objectives for themselves that go beyond monetary gain and emphasize the performance and growth of the individual, the community, and the environment. Sustainable human resource management (SHRM) is one of the fields that encourage the development of "green" businesses. The adoption of new ecological practices by an organization's human capital and the incorporation of cutting-edge sustainable strategies lead to the accomplishment of the sustainable development goals (SDGs) [19,20].

In this context, the primary aim of this research study is to empirically demonstrate the role of SHRM in attaining SDG 8 (decent work) and SDG 12 (responsible consumption and production), which is lacking in the existing literature on SDGs [20]. The empirical evidence can be used by organizations in setting different benchmarks, devising policies, visions and strategies. The organizations must embed changes at different hierarchical levels to achieve successful SDGs [21]. Furthermore, the study aims to investigate the role of green attitudes and green behavior as mediators between GHRM practices and SDGs, based on the theory of planned behavior which is an untouched area and leaves a significant gap in the literature [20]. The theory of planned behavior posits attitudes which build the behavior of a person. Recent examples in the relevant literature have emphasized that values play a

vital role in public communication and are a significant source in understanding individuals' behavior and attitudes [22]. There are two significant theories: first, the value–belief–norm (VBN) theory [23] and the supplies–values fit theory [24] support and explain the ways in which the values of individuals affect their behaviors. The VBN theory put forward personal values, beliefs, norms that will likely affect employee work behavior, and the authors in [25,26] have conducted empirical studies to understand the relationship between values and behaviors and reported a significant impact of personal environmental values on individual's environmentally friendly behavior. The present study provides a unique combination of relationships between GHRM and SDGs, which is missing in the previous literature to date [26]. It contributes to the literature on GHRM and SDGs by empirically examining a theoretical framework developed based on the theory of planned behavior (TPB). Consequently, the following research questions will be answered:

1. What is the impact of green HRM practices on SDGs, namely decent work and sustainable consumption behavior, in the manufacturing industry?
2. Is the relationship between green HRM and SDGs (decent work and sustainable consumption behavior) mediated through green attitudes and employee green skills in the manufacturing industry?
3. Do individual values act as moderators between green HRM and green attitudes as well as between green attitudes and sustainable consumption behavior in the manufacturing industry?

## 2. Theory and Hypothesis

### 2.1. Theory of Planned Behavior

The SHRM has been examined through various theoretical lenses in the literature, such as the stakeholder theory [26] (institutional theory [26,27], organizational development theory [28], the resource-based view [29] and the signaling theory [30]. The most widely used in the literature is the ability, motivation, and opportunity AMO theory, which not only provides a conceptual framework but also simplifies the strategies which promote SHRM [31,32]. In this context, past studies utilized the lens of AMO theory, to explain the relationship between human resources, and ethical, social and environmental performance. AMO, as a multi-dimensional model, promotes the sustainability of the firm at various levels, such as engagement and capacity-building, to promote green activities within and outside the organization to fabricate an eco-friendly atmosphere. In addition, it also includes the awareness of motivation for social activities as the responsibility of both the personnel and the firm. The duty of personnel is to show greater commitment to the sustainable procedures and the organization must compensate its employees when committed to sustainable practices [33]. Furthermore, the firm must provide a good working environment for its employees and shape an organizational culture that plays an important role in promoting a green attitude [32].

In the present research, the researchers have used the theory of planned behavior (TPB) as an underpinning theory to support the research framework. Recent research by [34,35] reveals that the theory of planned behavior can be used to predict the behavior of employees, particularly the environmentally friendly behavior [36] which can be quite useful for the organization's willingness to implement green practices. TPB is a social psychological model proposed by [37], which posits that a person's intention to carry out behavior is the immediate antecedent of that behavior, which is shaped by the attitudes, subjective norms, and perceived behavioral controls present in the close environment. The work climate of any organization is an important factor that influences the attitude of the workforce, leading them towards environmentally friendly behavior [38]. If the organization has a green climate and employees observe their co-workers being involved in environmentally friendly activities, it will create a positive perception in the employees' minds and they will be motivated toward environmentally friendly behavior [39]. It can be explained with the help of the theory of planned behavior which emphasizes that when an individual observes environmentally friendly activities, they develop a positive attitude

that inspires them toward environmentally friendly behavior [40]. The organization's increased focus on environmentalism and co-workers being involved in environmental activities can have a significant impact on shaping employee perception regarding green practices [38]. In addition, GHRM can be regarded as an important predictor of pro-environmental behavior in an organization. When organizations emphasize the importance of GHRM and communicate GHRM practices with more clarity to their workforce, this can help in developing pro-environmental behavior among employees and facilitate green initiatives in the workplace [13,41,42]. However, the organizations must engage employees in green initiatives through green job design and green tasks, and must compensate them for green achievements, which will inspire them toward green behaviors [43]. Therefore, as indicated by [36], attitude, pressure, and controllability all significantly influence the behavior of employees, which helps the firms in the adoption of green HR practices. For today's firm, green HR efforts may be considered a planned, ongoing shift. The formulation and execution of a medium- to long-term sustainability plan is the first step in the majority of green HR practices [44]. Furthermore, businesses can benefit from green HR, firstly, by embracing them to build their reputation as a green employer and attract talented people, and secondly, by reducing product and labor costs (such as recycling) to encourage employees' eco-friendly behaviors.

### 2.2. Green HRM and Sustainable Development Goals (SDGs)

Several scholars have tried to define green HRM [45], describing GHRM as "HRM activities, which amplify the positive environmental consequences". GHRM alludes to the HRM facets of environmental management [32]. It has also been explained as human resource activities that lead to pragmatic environmental outcomes [45]. GHRM contemplates a firm's focus on environmental protective actions through its strategies and urges executives to become concerned about the processes by motivating the employees to participate in activities minimizing environmental pollution in the working area. The shifting interest of firms toward environmentally friendly business strategies are the core concerns of human resources that must be upgraded by expanding its horizons, including environmental management so that it can modify its essential HR functions [46]. The concept of being "green" is widely appreciated. According to [47], environmentalism at the corporate level refers to the recognition of environmental concerns and the integration of environmental issues into the strategic and decision-making process of firms. Victor [48] observes that environmentalism is receiving attention on a global scale, which emerges from specific settlements and contracts to resist climate changes [47,49–51] from the extreme pollution as a result of high-profile industrial accidents. There exists extensive research literature on green marketing [52], green accounting [53], green retailing [54] and green management in general [55] in the field of management.

In the current conditions of rapidly minimizing resources, the "greenway" of doing business is going to be the next competitive business advantage. Companies have now started thinking deliberately about the need to be green as a part of business. Several scholars define green management as the process by which companies participate in environmental management by establishing environmental strategies [56]. The notion of 'green management' is defined here as the process whereby companies manage the environment by developing ecological management strategies. There is a real need for companies to achieve an equilibrium between the inexorable industrial boom and the preservation of the natural environment to keep it available for our future generations [57]. The adoption of HR practices, including knowledge management, employee participation, employee training, recruitment and selection encouraging diversity, and leadership, needs be incorporated in firms for environmental development [58]. GHRM refers to practices enhancing green initiatives through growing employee awareness and commitment to the issues of ecological sustainability. The field of GHRM is significant in organizations as it participates in other functional areas of green management, green operations, green marketing, green supply chain management, green finance and accounting.

As compared to Millennium Development Goals (MDGs), SDGs demand more active involvement from all the sectors of society to prosper. The primary medium of any community is children, women and youth, workers, businesses and industry, farmers, non-governmental organizations, local authorities, trade unions, indigenous people, and a scientific and technological community that facilitates the United Nations' activities.

To accomplish the SDGs, human resource capabilities and product development or management systems should be enhanced. The authors of [59] explained GHRM as a new innovative concept where organizational personnel are encouraged to employ greener practices that have a deep penetrating affects in the daily routines of the organization [59]. Occasionally, it is a cumbersome process as new changes are not readily absorbed by workers, and management greatly fears encountering resistance that could sabotage the whole initiative. As a result, enterprises striving for a greener future may refer to GHRM as a process for innovation. The human resource department critically determines the workings of the firm. Therefore, the green concept must be incorporated into the human resource practices of any company. The environmental sustainability in the organization can be determined by the HR functions, by associating the people-related practices and policies with the sustainability goals that represent the eco-centered intentions of the firm.

Companies are now adapting their primary business strategies toward the new environmental-oriented schema, and HR should update its command and enlarge the scope by including environmental management to change the way it performs its vital HR functions. The authors in [46,60] recommend that HR has the capacity to evaluate and impact employee sustainability-related behaviors and attitudes. Such indicated learning and development are often referred to as human resource development (HRD).

### 2.2.1. GHRM and SDG 12 Sustainable Consumption and Production Behavior (SCPB)

Green HRM is a strategy for achieving environmental sustainability [61]. Green HRM is a multifaceted concept that encompasses a variety of best practices [31,62,63]. Employee participation and contributions via new ideas, similar values and objectives, environmental skills and expertise, formal and informal daily interactions, and decision-making are critical to generating a bottom-up and cross-functional process. Green HRM helps in the successful creation and implementation of company environmental policies by aligning training, selection, hiring, incentives, and performance evaluation with sustainability strategies [64].

Green recruiting and selection (GRS) are one of the human resource management tasks, according to [31,65]. Human resources managers have one of the most difficult challenges in the world: searching for and retaining professional staff [66]. Eco-conscious companies are promoting themselves in order to draw the attention of competent as well high-profile professionals who are anxious about implementing greener practices for a better future. Job seekers, on the other hand, may prepare for the international norms of green culture by becoming green workers. For their primary businesses, green employees favor companies that are environmentally and socially conscious [67].

It is also vital to identify the employees willing to participate in environmental management activities and volunteer. Green training and education extensive environmental employee training has a substantial impact on overall environmental sustainability [67]. Employees' understanding of the environmental impact of their employers' actions is increased through green education and training programs [68]. Employees are educated on environmental problems both intellectually and emotionally, and they are informed about potential answers to ongoing issues. Eco-friendly employees play a pivotal role by generating new ideas that are environmentally friendly [69]. Such employees also help in raising the eco-awareness level of other employees which aids the top management in the implementation of environmental strategies. Reward systems are commonly reported in the literature as a fuel for stimulating employees and fostering their dedication to environmental responsibilities [70,71]. The purpose of an incentive system is to draw in, keep, and motivate employees to achieve environmental goals. The authors in [32,72] claimed that

rewards and incentives might be the most successful method of aligning firm objectives with workers' self-interest goals.

The purpose of green assessment and performance management (PM) is to compare goals and outcomes to examine and evaluate employees' performance in relation to their duties and responsibilities [73]. Employees receive meaningful and constructive feedback on their contributions to environmental sustainability when PM is used for environmental issues. Feedback can help to prevent negative attitudes and reinforce positive conduct [74]. Furthermore, [75] asserted that rather than being stuck in the same green behaviors and skills, green appraisal must be dynamic and incorporate new goals and challenges. Environmental sustainability-oriented employee participation is critical for discovering possible green possibilities [32] and improving the most significant environmental sustainability outcomes.

**H1.** *Green HRM is positively associated with SDG 12 (Sustainable Consumption and Production Behavior (SCPB).*

### 2.2.2. Green HRM and SDG 8 (Decent Work)

The economic prosperity of individuals is the nuclei of SGDs which ensures a decent work environment and future growth of employees in the organization [76]. Decent work as defined by (ILO) is productive work for employees in the condition of freedom, equity, security, and human dignity. The authors in [77] emphasized that decent green jobs are a main solution and an indispensable key to building a sustainable and low-carbon global economy.

As emphasized in previous studies, green HRM plays a vital role in treating all employees equally. It encourages social integration and provides employees with social protection. Employees are free to demonstrate their concern, to organize, to offer prospects for personal development. It is pertinent to mention here that economic sustainability generates practices for economic development and decent work. The indicated sustainability is achieved through valued, efficient resources. Green HRM practices ensure sustainable economic prosperity, labor productivity, employee opportunity, and a secured working environment [78,79]. Therefore, the second hypothesis is:

**H2.** *Green HRM is positively associated with SDG 8 (Decent work).*

### 2.2.3. GHRM and Employee Attitudes

In recent times, scholars have considered GHRM practices as the key element in attaining environmental management [80], due to their deeper impact on employee outcomes [81]. The authors of [66,82] emphasized that HRM practices enhance employee attitudes towards the organization. In a recent study, [83] concluded that green HRM practices improve the attitudes of employees related to their employer, and attitudes towards environment-related initiatives can also be influenced [84]. Increased competition, challenges and complication in HRM have introduced and enhanced the strategic HRM practice in management. According to the P–E fit perspective, SHRM is related to employee values, behavior and knowledge, meaning that it ultimately enhances positive employee attitudes [85]. Hence, strategic HRM practices help to influence and elevate employees' attitudes, such as commitment. Green human resource practices affect the environmental behavior and ecological performance of the employees [86]. SHRM is willing to recognize the work of employees and ready to contribute to their working process. As such, treasurable investment, recognition and encouragement increases the motivation of employees to embrace and adopt positive attitudes and productive behavior in the organization [87]. The perception of employees is very strongly related to the attitude of the employees' performance and their behavior as well [88]. Hence, the third hypothesis is stated as:

**H3.** *GHRM is positively associated with Green Attitude.*

### 2.2.4. GHRM and Employee Skills

"Ecological consciousness" is the new buzzword sweeping the workplace [57]. Laborers contend that to carry out a successful corporate green management framework, advancing a lot of specialized skills and broad abilities among all the employees of the organization must be part and parcel of the organization strategy as it provides a firm with a competitive advantage. The authors of [89] emphasized that training and empowerment programs in the organization provide new skills and competences among the employees of "pro green" organizations. The authors of [90] empirically demonstrated that green performance and appraisal are important factors in the GHRM, encouraging workers to improve their professional abilities to meet company goals. Furthermore, to prevent environmental degradation, GHRM emphasizes the improvement of employees' skills and knowledge through training in energy conservation, waste reduction, environmental awareness dissemination, and opportunities for employees to participate in environmental problem solving. GHRM provides a more fertile ground for employee training and development [91]. Employee green skills are nourished when their green ideas are welcomed by the management, and this encourages a sense of responsibility towards sustainability among employees [92]. Therefore, the fourth hypothesis is proposed as:

**H4.** *GHRM is positively associated with Employee Skills.*

### 2.2.5. Green Attitudes and SDG 12 (Sustainable Consumption and Production Behavior (SCPB)

Green behavior (GB) is linked with the "humanistic conduct" which a person uses to deal with colleagues at the workplace, with the firm as a whole, with the public and social communities, and with the environment. These attitudes are professed as "good" deeds that value "collective" concerns. A conceptual framework proposed by [39] helps to observe two types of green behavior of employees (EGB), required EGB and voluntary EBG. The green employee behavior that is executed while working is the "required EGB", it is also described as task-related EGB. On the other hand, the voluntary EGB is more like organizational citizenship behavior, including social and personal initiatives in correspondence with the external and internal work environment consisting of tasks other than the firm's millennium needs. Norton et al.'s work concluded with a framework originating from "person-environment interaction, taxonomy of job performance, and self-determination theory" [93].

The nature of the green behavior of an employee is "pro-social", and in a realistic view, the green behavior of an employee at the workplace consists of in-role behavior and extra-role green behaviors. The authors of [67] state that both of these dimensions of green behaviors positively influence the organizational outgrowth by value creation, and how that behavior is defined, whether in-role or extra-role, depends upon the organization and its expectation regarding their employees [94]. There is a possibility of a demand for green behavior in various jobs, as there are a number of jobs that require employees to make sure that polluted or toxic water should not be poured in the drinking water systems, and many more besides. The indicated workplace behavior is considered mandatory and thus treated as part of an employee's primary duties. The extra-role green behavior is less-well defined, and can be treated as a suggestion to improve the business's environmental performance, an example of which would include keeping the lights and computers off when not in use [94]. The in-role and extra-role behaviors are considered salient for achieving organizational "green goals" [39]. There is a possibility of different antecedents because employees possess different degrees of discretion about when and how to show these behaviors in the workplace [95].

In the normative conduct approach, [96] suggested that human behavior is guided through norms as they emphasize the social outcomes of participating in particular activities. Socially acceptable conduct is at the heart of these theories. Research on sustainability which is based on this theory is mainly concentrated on green (pro-environmental) behavior

(in private) (for further explanation, see [96]). The authors in [39] have tried to examine the perception of employees' organizational norms to explain the green behavior of employees. The role of interaction, especially the interaction between individuals and some other party (entity), for example, leaders and groups, are the main focus of exchange theories [97]. Recently, the notion of social exchange has been used to better understand environmental citizenship practices [94,98].

Th authors in [99] have hypothesized, based on this perspective, that reciprocity (interaction) between employees and their organization may play the role of a mediator in the relationship between the environmental attitudes and environmental citizenship behaviors of employees. Theories of motivation focus on what makes someone choose to engage in a certain action. For instance, according to [100], the theory of self-determination states that autonomous and controlled motivations result in behavior. An employee feels satisfaction by performing an activity, then he/she is motivated to be engaged in that activity or behavior (such as green employee behavior), just as if they believe that they will be rewarded by the firm (controlled motivation). Self-determination theory [101] justifies and explains the EGB as having multiple motivators combined with attitudes and values. Nowadays, people are showing more care and are more concerned about the planet and ecological lifestyle. The authors in [102] determined that there is a negative correlation between inequality and environmental behavior, which means that increased environmental behavior discourages the level of inequality. Humans are the primary cause of global change in the climate; different organizations can play their role in achieving sustainable development goals by triggering the sustainable behavior, attitudes and behaviors of employees working in these organizations [103]. Adapting the strategies which support sustainable development goals requires this change in behavior [104]. Many behavioral assumptions are the basis of all strategies which help to attain pathways to better consumption and production [16,105]. Enacting in-depth behavioral steps is essential to achieve sustainable development goals such as lower inequality, more productive consumption and less wastage, more recycling (switch off the lights when leaving the office, recyclable waste) and decent work [103]. Organizations are now trying hard to change their employees' behaviors to address and solve many issues such as recycling, reducing the emission of greenhouse gases (GHG) and reducing the use of energy and water [106]. The authors in [107] alleged that behavioral change also helps to mitigate the issue of climate change and other environmental problems, including biodiversity loss. Therefore, organizations are focusing on increasing the environmental behavior of their employees. The following hypothesis was developed based on the literature discussed above:

**H5.** *Green Attitude is positively associated with Sustainable Consumption and Production Behavior.*

### 2.2.6. Green Attitude and Decent Work

When employees have a green mindset, the company's macro-level sustainable development plan may be translated into actual practice at the micro level [108]. Environmentally conscious workplace conduct is advantageous to both businesses and society as a whole. The present and future generations may benefit from environmentally responsible activities [109]. Decent and harmonious employment depends on decent labor, but this is not always the case in global industrial networks. Lack of empirical evidence is a hurdle to understanding the key relationship between green employee attitude and decent work. According to the study conducted by [98], Chinese firms reported a positive association between employee green attitude and decent work. The study concluded that garment manufacturers are seeing a shift in the importance of non-cognitive criteria, such as how employees feel about having a job they can be proud of. The sixth hypothesis is projected as:

**H6.** *Employee Green Attitude is positively associated with Sustainable Development Goal 8 (Decent Work).*

### 2.2.7. Employee Skills and Sustainable Consumption and Production Behavior (SCPB)

Design, manufacturing, administration, control over technologies, and technical know-how all fall under the umbrella of "green skills," according to a recent study [110,111]. Regulation of the environment promotes technological advancement and raises the need for people with technical and scientific backgrounds, 140. Hence, development of employee skills (green), in line with the corporate green strategy, is deeply rooted in the employee attitude that leads an employee to initially develop such skills and then behave in a sustainably responsible manner by practicing SCP. The authors of [112], in a comparative analysis, concluded that employees recognize "green skills" as eco-friendly and have no proper understanding of the role of "green skill", which is big hurdle towards the adoption of a SCPB. The authors of [113] argued that companies should provide a more supportive working environment (supportive organizational and material structures) where employees could use their green skills and experiment with sustainable consumption practices, while the authors of [114] theoretically demonstrated that collaborative learning in organizations nourishes specific pro-environmental skills among employees that make them ethically responsible for sustainable consumption. Therefore, the next working hypothesis is proposed as:

**H7.** *Employee Skills is positively associated with Sustainable Consumption and Production Behavior (SCPB).*

### 2.3. Mediation Hypotheses Development

Earlier studies indicated the importance of attitudes and behaviors as critical mediators between HRM and sustainability. The authors of [115] attempted to examine the mediation of reactions between HRM practices and employee behavior and provided evidence that employee attitudes are the potential mediators in the HRM–employee behavior relationship. Based on social exchange, they posited that when the organization supports its employees, their values can be translated into positive desired employee behavioral outcomes. Attitudes that emerge from the presence and implementation of high-performance HR systems play an essential role in producing desired behaviors [87]. Furthermore, [116] asserted that work attitudes are the potential mediator between HRM practices and organizational outcomes. Moreover, it is essential to guide and transform human attitudes to initiate pro-environment behaviors [117], which will help to achieve sustainable development [117].

**H8.** *Employee Green Attitude mediates the relationship between GHRM and SDG 12 (SCB).*

**H9.** *Employee Green Attitude mediates the relationship between GHRM and SDGs 8 (Decent Work).*

### 2.4. Moderating Effect of Individual Green Values

Green behavior (GB) is linked with the "humanistic conduct" a person uses to deal with colleagues at the workplace, with the firm as a whole, with the public and social communities, and with the environment. These attitudes are professed as "good" deeds that value "collective" concerns. A conceptual framework proposed by Norton et al. (2015) helps to observe two types of green behavior in employees (EGB): required EGB and voluntary EBG. The green employee behavior that is executed while working is the "required EGB", which is also referred to as task-related EGB. On the other hand, the voluntary EGB is more akin to organizational citizenship behavior, including social and personal initiatives in correspondence with the external and internal work environment consisting of tasks other than firm's millennium needs. Norton et al.'s work concluded with a framework which originated from "person-environment interaction, taxonomy of job performance, and self-determination theory" [31].

The nature of the green behavior of an employee is "pro-social", and in a realistic view, the green behavior of an employee at the workplace consists of in-role behavior and extra-role green behaviors. The authors of [67] state that both of these dimensions of green

behaviors positively influence the organizational outgrowth by value creation, and how that behavior is defined, whether in-role or extra-role, depends upon the organization and its expectation regarding their employees [94]. There is a possibility of a demand for green behavior in various jobs, as there are a number of jobs that require employees to make sure that polluted or toxic water should not be poured in the drinking water systems, and many more besides. The indicated workplace behavior is considered mandatory and thus treated as part of an employee's primary duties. The extra-role green behavior is less-well defined, and can be treated as a suggestion to improve the business's environmental performance, an example of which would include keeping the lights and computers off when not in use [94]. The in-role and extra-role behaviors are considered salient for achieving organizational "green goals" [39]. There is a possibility of different antecedents because employees possess different degrees of discretion about when and how to show these behaviors in the workplace [95].

In the normative conduct approach, [96] suggested that human behavior is guided through norms as they emphasize the social outcomes of participating in particular activities. Socially acceptable conduct is at the heart of these theories. Research on sustainability which is based on this theory is mainly concentrated on green (pro-environmental) behavior (in private) (for further explanation, see [96]). The authors in [39] have tried to examine the perception of employees' organizational norms to explain the green behavior of employees. The role of interaction, especially the interaction between individuals and some other party (entity), for example, leaders and groups, are the main focus of exchange theories [97]. Recently, the notion of social exchange has been used to better understand environmental citizenship practices [94,98].

The authors in [99] postulated that reciprocity (interaction) between employees and their organization may play the role of mediator in the relationship between environmental attitudes and environmental citizenship behavior of employees. Motivation theories are built around the factors that drive the decision to be involved in a particular behavior. For instance, the theory of self-determination, [100] as autonomous and controlled motivations result in behavior. According to this view, if an employee feels satisfaction by doing an activity, then he/she is motivated to be engaged in that activity or behavior (such as green employee behavior) or if they think that the company will reward them (controlled motivation). The authors of [101] have used self-determination theory to support and explain EGB as having different motivators alongside, such as attitudes and values. Nowadays, people are showing more care and are more concerned about the planet and ecological lifestyle [101]. A study conducted by [102], concluded that there is a negative correlation between inequality and environmental behavior, which means increased environmental behavior discourages the level of inequality. Humans are the primary cause of global change in the climate; different organizations can play their role in achieving sustainable development goals by triggering the sustainable behavior, attitudes and behaviors of employees working in these organizations [103]. Adapting the strategies which support sustainable development goals require behavioral change [104]. Numerous behavioral assumptions are the basis of all strategies which help to attain pathways of better consumption and production [16,105]. Taking more thorough behavioral steps rather than just CSR is essential to achieve sustainable development goals such as lower inequality, more productive consumption and less wastage, more recycling (switching off the lights when leaving the office, recyclable waste), and decent work [103]. Nowadays, organizations are trying hard to change their employees' behavior to address and solve many issues like recycling, reducing the emission of greenhouse gases (GHG), and reducing the use of energy and water [106]. [107] alleged that behavioral change also helps to mitigate the issue of climate change and other environmental problems such as biodiversity loss. Therefore, organizations are focusing on increasing the environmental behavior of their employees. The following hypotheses were developed based on the above-discussed literature, value-belief-norm [25] and theory of supplies-values fit [118], this study develops the following hypotheses:

**H10.** *Individual Values positively moderate the relationship between GHRM and Employee Green Attitudes.*

**H11.** *Individual Values positively moderate the relationship between Green Attitudes and SDG 12 (Sustainable Consumption and Production).*

*2.5. Research Framework*

The present research's framework is based on the literature studied above to explore the link between GHRM practices, green attitude skills and sustainable practices. The links between GHRM practices, green attitude, green skills and sustainable practices are illustrated in Figure 1.

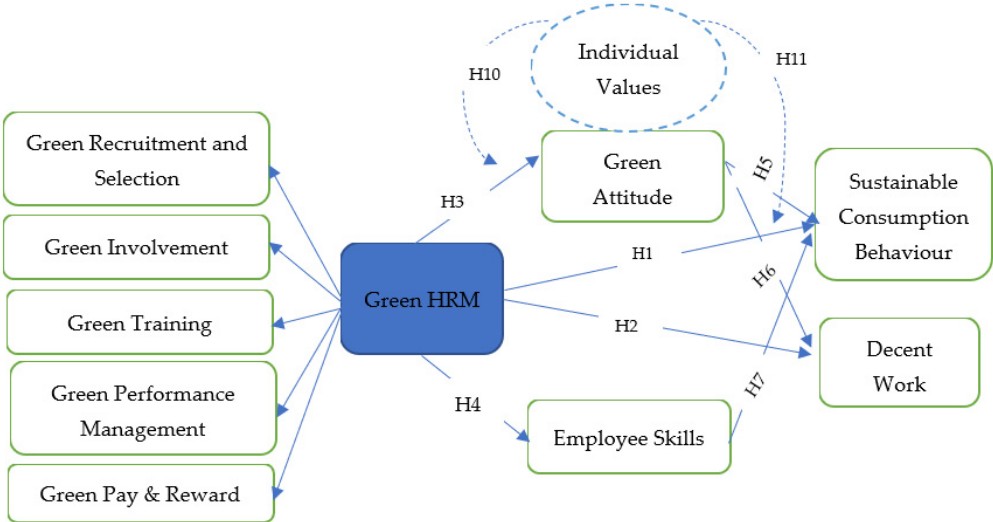

**Figure 1.** Link between GHRM practices, green attitudes, employee skills and sustainable behavior.

**3. Methodology**

*3.1. Instrument Design*

The researchers collected the data via the use of a survey questionnaire. The questions on the survey were created with ease of use in mind to truly reflect the model's elements. It is noteworthy that the questionnaire was prepared in conformity with the theoretical explanations in the literature. The justification can be traced to the work [119,120]. A 5-point Likert scale was used to access the indicators. All the indicators used in the study were reflective. Items for measuring GHRM were mainly adopted from [63]. Green attitude measurement was adopted from [121]. Measurement of green skills was adopted from [122]. Sustainable consumption behavior was adopted from [123] and the decent work measure was adopted from [124].

*3.2. Sample of the Study*

Positivism research philosophy is considered to be suitable to conduct this research as the current study was performed in a natural setting. Hence, a quantitative research method is used and a deductive approach is adopted because it progresses from theory to test hypothesis and confirmation of results. The data were collected through survey questionnaires, meaning that the study is related to survey research.

The study's target population was Pakistani hi-tech manufacturing units. Recently, the government and business have teamed up to boost exports and take specific measures to ensure their long-term viability. A total of 465 enterprises were selected from a list as furnished by the Pakistani government's Ministry of Industry Division. The authors in [125] devised a table to help calculate the sample size, based on which 214 is the required sample size. Top-level executives such as general managers and HR managers, etc., who were

part of policy/strategy development at those organizations, were the respondents of the study. Data were collected using simple random sampling. Furthermore, in a probability sampling strategy, a representative sample is critical for generalization reasons. This study uses a simple random sample to ensure a fair and independent display of the data. In total, 231 of the 465 surveys sent out were returned, giving a response rate of over 49.6%. Only 197 useable completed surveys were accepted for further research out of a total of 231 returned surveys, resulting in a usable response rate of 42.3%. All who responded were well-trained professionals with bachelor's and master's degrees. In addition, they had abundant experience in the current company in developing strategies related to green skills and environmental awareness, and were considered to be suitable for this research. Smart Pls 3 was used to analyze the data.

### 3.3. Measurement Model

CFA (confirmatory factor analysis) has been applied in this research. The loadings were found to be above 0.5 (more than 50%) [126] as shown in Table 1. CR values ranged between 0.808 and 0.904 [127]. F and L Criterion was also established, as shown in Table 2, and HTMT value is below 0.90, as shown in Table 3, thus discriminant validity holds [126]. The VIF values were found to be less than 2 as per criterion of [126]. Hence, the measurement model shows a good fit.

**Table 1.** Loadings, AVE, Cronbach's alpha and CR.

| | | Factor Loadings | Composite Reliability | Average Variance Extracted (AVE) |
|---|---|---|---|---|
| Green HRM (Second Order Construct) | | | 0.904 | 0.578 |
| | Green Recruitment and Selection | | 0.81 | 0.517 |
| | GH1 | 0.789 | | |
| | GH2 | 0.686 | | |
| | GH3 | 0.802 | | |
| | Green Involvement | | 0.843 | 0.574 |
| | GI1 | 0.789 | | |
| | GI2 | 0.798 | | |
| | GI4 | 0.697 | | |
| | GI6 | 0.743 | | |
| | Green Training | | 0.82 | 0.603 |
| | GT1 | 0.738 | | |
| | GT2 | 0.78 | | |
| | GT3 | 0.809 | | |
| | Green Performance Management | | 0.81 | 0.517 |
| | GP1 | 0.75 | | |
| | GP2 | 0.698 | | |
| | GP3 | 0.719 | | |
| | GP4 | 0.706 | | |
| | Green Pay and Reward | | 0.808 | 0.586 |
| | GPR2 | 0.792 | | |
| | GPR3 | 0.816 | | |
| | GPR4 | 0.701 | | |
| Employee Green Attitude | | | 0.87 | 0.61 |
| | GA1 | 0.722 | | |
| | GA2 | 0.83 | | |
| | GA3 | 0.78 | | |
| | GA4 | 0.784 | | |

**Table 1.** *Cont.*

| | | Factor Loadings | Composite Reliability | Average Variance Extracted (AVE) |
|---|---|---|---|---|
| | GA5 | 0.785 | | |
| Individual Values | | | | |
| | INV1 | 0.776 | 0.833 | 0.556 |
| | INV2 | 0.75 | | |
| | INV3 | 0.741 | | |
| | INV4 | 0.713 | | |
| Employee Skills | | | 0.839 | 0.636 |
| | ES4 | 0.85 | | |
| | ES5 | 0.83 | | |
| | ES6 | 0.78 | | |
| Decent Work | | | 0.91 | 0.629 |
| | DC1 | 0.855 | | |
| | DC2 | 0.802 | | |
| | DC3 | 0.84 | | |
| | DC4 | 0.721 | | |
| | DC5 | 0.795 | | |
| | DC6 | 0.74 | | |
| Sustainable Consumption and Production Behavior | | | 0.913 | 0.679 |
| | SCB1 | 0.888 | | |
| | SCB2 | 0.873 | | |
| | SCB3 | 0.85 | | |
| | SCB4 | 0.801 | | |
| | SCB5 | 0.705 | | |

**Table 2.** Fornell and Larker Criterion for Discriminant Validity.

| | Decent Work | Employee Skills | Employee Green Attitude | Green Pay and Reward | Green Performance Management | Green Training | Green Recruitment and Selection | Green Involvement | Individual Values | Sustainable Consumption and Production Behavior |
|---|---|---|---|---|---|---|---|---|---|---|
| Decent Work | 0.793 | | | | | | | | | |
| Employee Skills | 0.669 | 0.797 | | | | | | | | |
| Employee Green Attitude | 0.637 | 0.663 | 0.776 | | | | | | | |
| Green Pay and Reward | 0.535 | 0.577 | 0.655 | 0.765 | | | | | | |
| Green Performance Management | 0.551 | 0.509 | 0.585 | 0.52 | 0.765 | | | | | |
| Green Training | 0.553 | 0.516 | 0.469 | 0.536 | 0.652 | 0.73 | | | | |
| Green Recruitment and Selection | 0.461 | 0.471 | 0.428 | 0.507 | 0.474 | 0.538 | 0.797 | | | |
| Green Involvement | 0.653 | 0.614 | 0.639 | 0.64 | 0.608 | 0.595 | 0.489 | 0.758 | | |
| Individual Values | 0.716 | 0.639 | 0.741 | 0.608 | 0.625 | 0.6 | 0.552 | 0.69 | 0.75 | |
| Sustainable Consumption and Production Behavior | 0.658 | 0.598 | 0.527 | 0.359 | 0.442 | 0.469 | 0.195 | 0.463 | 0.5 | 0.824 |

**Table 3.** HTMT Criterion for discriminant validity.

| | Decent Work | Employee Skills | Green Attitude | Green Pay and Reward | Green Performance Management | Green Training | Green Recruitment and Selection | Green Involvement | Individual Values | Sustainable Consumption Behavior |
|---|---|---|---|---|---|---|---|---|---|---|
| Decent Work | | | | | | | | | | |
| Employee Skills | 0.841 | | | | | | | | | |
| Green Attitude | 0.672 | 0.786 | | | | | | | | |
| Green Pay and Reward | 0.718 | 0.863 | 0.878 | | | | | | | |
| Green Performance Management | 0.703 | 0.73 | 0.787 | 0.764 | | | | | | |
| Green Training | 0.712 | 0.738 | 0.596 | 0.806 | 0.845 | | | | | |
| Green Recruitment and Selection | 0.602 | 0.708 | 0.592 | 0.754 | 0.704 | 0.816 | | | | |
| Green Involvement | 0.801 | 0.841 | 0.809 | 0.913 | 0.828 | 0.843 | 0.69 | | | |
| Individual Values | 0.885 | 0.889 | 0.774 | 0.916 | 0.843 | 0.793 | 0.806 | 0.834 | | |
| Sustainable Consumption Behavior | 0.738 | 0.733 | 0.544 | 0.48 | 0.562 | 0.589 | 0.259 | 0.557 | 0.612 | |

### 3.4. The Assessment of the Inner Model and Hypotheses Testing Procedures

As evident in Table 4, the direct effect of GHRM on employee green attitude, employee skills, decent work and sustainable consumption behavior was found to be ($\beta = 0.700$, $p < 0.000$), ($\beta = 0.784$, $p < 0.000$), ($\beta = 0.297$, $p < 0.003$), and ($\beta = 0.410$, $p < 0.000$), respectively. Similarly, the direct effect of employee skills on sustainable consumption behavior was found to be ($\beta = 0.411$, $p < 0.000$), while the effect of green attitude on decent work and SCPB was found to be $\beta = 0.297$, $p < 0.003$) and ($\beta = 0.410$, $p < 0.000$).

**Table 4.** Structural model (direct and indirect effects).

| Direct Effects | Beta | t-Values | *p*-Values | Hypothesis | Beta | t-Values | *p*-Values | Indirect Effects Hypothesis |
|---|---|---|---|---|---|---|---|---|
| Employee Skills–Sustainable Consumption and Production Behavior | 0.411 | 3.239 | 0.001 | Accepted | | | | |
| GHRM–Decent Work | 0.207 | 2.586 | 0.010 | Accepted | | | | |
| GHRM–Green Attitude | 0.700 | 18.72 | 0.000 | Accepted | | | | |
| GHRM–Sustainable Consumption and Behavior | 0.091 | 1.425 | 0.463 | Rejected | | | | |
| GHRM–Employee Skills | 0.784 | 19.51 | 0.000 | Accepted | | | | |
| Employee Green Attitude–Decent Work | 0.297 | 2.935 | 0.003 | Accepted | | | | |
| Employee Green Attitude–Sustainable Consumption and Production Behavior | 0.410 | 4.107 | 0.000 | Accepted | | | | |
| GHRM–Sustainable Consumption and Production Behavior Specific Indirect Effects | 0.091 | 0.735 | 0.463 | Rejected | | | | |

**Table 4.** *Cont.*

| Direct Effects | Beta | t-Values | p-Values | Hypothesis | Beta | t-Values | p-Values | Indirect Effects Hypothesis |
|---|---|---|---|---|---|---|---|---|
| GHRM–Employee Green Attitude–Decent Work | | | | | 0.208 | 2.586 | 0.010 | Accepted |
| GHRM–Employee Skills–Sustainable Consumption and Production Behavior | | | | | 0.273 | 3.307 | 0.001 | Accepted |
| GHRM–Employee Green Attitude–Sustainable Consumption and Production Behavior | | | | | 0.167 | 2.046 | 0.041 | Accepted |

The indirect effects are concerned the indirect impact of GHRM–green attitude–decent work was found to be ($\beta = 0.208$, $p < 0.010$) while the direct path from GHRM to decent work was ($\beta = 0.207$, $p < 0.010$). Thus, green attitude is shown to be a complementary mediation (partial mediation) between GHRM and decent work. GHRM–Employee Skills–SCPB was ($\beta = 0.2738$, $p < 0.001$); while the direct path of GHRM to sustainable consumption behavior was found to be insignificant ($\beta = 0.091$, $p < 0.463$). Thus, employee skills is shown as having only an indirect mediation (full mediation) between GHRM and sustainable consumption behavior. GHRM–Employee Attitude–SCPB was ($\beta = 0.167$, $p < 0.041$).

## 4. Moderation Analysis

The results of the structural analysis in Smart PLS 3 showed the moderating effect of individual values between GHRM and green attitude ($0.067$, $p < 0.05$) as shown in Table 4. The authors in [128] recommended drawing an interaction plot to further access the moderating effect. It could be inferred from Figure 2 that the positive impact of the GHRM and green attitude becomes stronger with a higher level of individual values. Thus, our proposed hypothesis is reinforced.

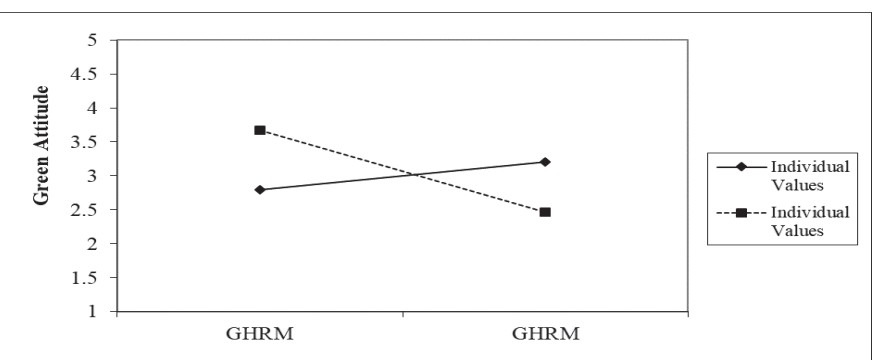

**Figure 2.** Moderator analysis of Green Attitude and GHRM.

The results of the structural analysis in Smart PLS 3 showed the moderating effect of individual values between green attitude and SCBP ($0.082$, $p < 0.05$) as shown in Table 5. The authors in [128] recommended drawing an interaction plot to further access the moderating effect. It could be inferred from Figure 3 that the positive impact of the green attitude and SCPB becomes stronger with a higher level of individual values. Thus, our proposed hypothesis is reinforced.

**Table 5.** Results of moderator analysis.

| Construct | Path Coefficient (β) | T-Statistics |
| --- | --- | --- |
| Moderating effect 1–Employee Green Attitude | 0.067 | 1.866 |
| Moderating effect 2–Sustainable Consumption and Production Behavior | 0.082 | 2.56 |

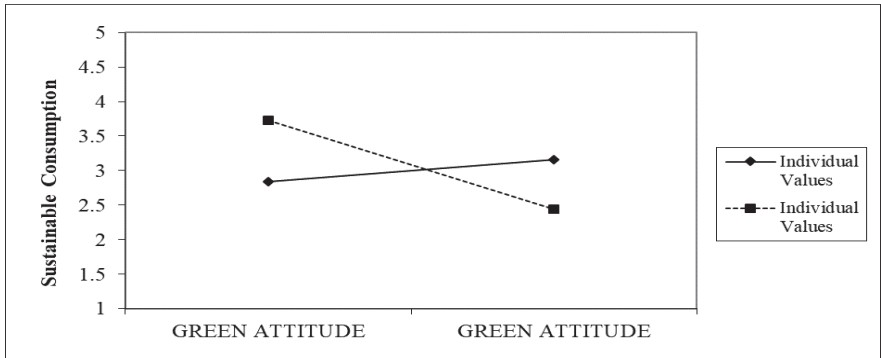

**Figure 3.** Moderator Analysis of Sustainable Consumption and Green Attitude.

## 5. Discussion and Conclusions

The empirical evidence of the current study identifies the contribution of GHRM practices to the attainment of SDGs. It reviews how the implementation of GHRM practices such as green recruitment and selection, green involvement, green training, green performance management, green pay, and reward lead to the achievement of SDGs. As illustrated in Figure 2, GHRM has a significant positive effect on employees' skills. The GHRM practices create awareness among employees regarding green behavior and also provide them with the necessary skills required for the task [129]. Similarly, according to various scholars such as [81,84], the GHRM has a significant positive effect on the green attitude. It plays an important role in shaping environmentally friendly behavior among employees and taking environmentally friendly initiatives for the well-being of the organization and society. A green attitude among employees has a significant positive effect on shaping green environmental behavior among employees, as is evident from the existing literature which suggests that pro environmental attitudes positively predict green behavior among employees [130]. Green behavior is positively associated with SDGs and also mediates the relationship between green attitudes and SDGs. The existing literature suggests that sustainable goals are achieved by triggering changes in the behavior of employees [103,104]. Individual values positively moderate the relationship between GHRM and green attitudes. Values vary from person to person, if the values of employees correspond more with the organization's values it will be easier for the organization to incorporate GHRM practices and develop green attitudes among the employees [24,131]. Individual values also moderate the relationship between green attitudes and green behavior. Therefore, if an organization provides a favorable environment for employee values (green practices for green values), then there will be harmony between employee and organizational values, and it is expected that employees will exhibit more environmentally friendly behavior and take green initiatives [25]. Legitimacy, competitiveness, and environmental responsibility are the motives that are required to encourage organizational change [68]. The revolution is described as a paradigm shift toward "green" management [132,133]. The contribution of the present study to the literature is the investigation of the impacts of green human resource management on SDG 8 (decent work) and SDG 12 (sustainable consumption and production) in the industry. It utilized a mediation model where green attitudes and green skills serve as mediators in the relationship between green HRM components (green training, green recruitment and selection, green involvement, green performance management

and green pay and reward), and sustainable consumption and production and decent work. The results are in line with the existing literature as it identifies that GHRM practices are beneficial and help in achieving sustainable development goals.

### 5.1. Theoretical Contribution

Firstly, the current study focused on a large-scale green manufacturing environment in a non-Western nation, expanding the literature on GHRM and SCPB as well as GHRM and decent work. The industry's top management and employees are better able to grasp these two essential principles in the workplace. The mediating influence of employee attitude (green) and skill (green skills) between GHRM and SCPB, as well as between GHRM and decent work, is also unknown.

Secondly, the study empirically examined the inter-relationships among constructs of GHRM, green attitude, employee skills, SCPB, and decent work in Pakistan's textile setting. The existing knowledge level in relation to the key components investigated is, therefore, extended. This study highlights the importance and impact of GHRM integrated with green attitudes and employee skills delivered directly by managers in bringing out and appealing to the SCPB in the textile industry based upon the TPB theory.

Thirdly, TPB theory may be applied to a non-Western setting in order to better understand the influence of manager–subordinate relationships on SCPB as well as a decent working environment, notably among employees in Pakistani textile manufacturing firms. Thirdly, this study has shifted away from the old method of assessing green HRM in Pakistan's textile manufacturing business via a one-dimensional approach. As an alternative, it is likely that the conceptualization of GHRM which follows [63] provides a more accurate reflection of organizational GHRM practices. The amount of information collected through a multi-dimensional construct presents a better and more accurate picture of that construct in comparison to a uni-dimensional construct [134].

Fourth, the study findings regarding the mediating role of employee's green attitudes might be seen as novel insights, as this research is one of the few examples in the literature on green attitude towards SCPB as well as decent work environment. Being mindful of the popularity of research conducted in the West, this research is one of the few studies on green attitude and employee skills conducted in an emerging economy, and we show our model's potential value for understanding the GHRM and SCPB relationship via a key mechanism, green employee attitude and employee skills, which until today, has been lacking in the literature, as identified by [20].

Lastly, the research contributes to the literature on SDGs in terms of the mechanism through which organizations can achieve SDG-related goals through HRM practices. Green HRM is considered a critical area that monitors the use of natural resources and introduces sustainable development goals in all organizational areas [135]. Moreover, HRM is a human-centered approach that leaves behind firms' traditional outlooks which mainly focused on reducing costs and maximizing economic output [136]. Green HRM considers the influence of internal as well as external factors, such as governmental and community pressures, social and ecological policies and rules, the needs of consumers, and the welfare of employees [137]. Green HRM includes five significant practices: green training, green recruitment and selection, green involvement, green performance management, and green pay and reward. These green practices ensure the efficient and effective management, allocation, and consumption of natural resources and encourage a certain level of awareness and responsibility among individuals working in the organization.

### 5.2. Practical Contribution

At the organizational level, green characteristics consist of ecological behavior, green competencies of employees, and green values. These characteristics are perceived to be the key drivers of sustainable performance. Once the employees working in the organization develop a green attitude and green skills, then it becomes more feasible for the organization to achieve SDGs. While at the organizational level, the antecedents of sustainability include:

supportive culture, promoting collectivistic corporate identity and implementation of green HR functions. Nowadays, it is much easier to understand that when management practices green HRM in the organization, it contributes to enhancing the green competencies of employees, which ultimately results in the attainment of SDGs. The findings of the study suggest that green HRM has a significant positive impact on the sustainable consumption behavior, and a green attitude and green skills mediate this effect. Past studies also authenticated the findings of the present study, as [138] concluded that human resource management positively impacts the attitudes of employees working in the organization, and it also impacts the green skills of an employee by providing sufficient rewards to employees showing their green skills at the workplace. The authors of [33] also found that employee green inputs (green attitude and green skills) contribute to enhancing the green behavior of employees, which results in green organizational performance, while meeting all corporate social responsibility (CSR) requirements [139]. The researchers also found that green HRM has a significant positive relationship with decent work and this relationship is mediated by a green attitude. These results are similar to previous findings [20,138], which show that when the management of an organization implements GHRM practices, it boosts the green attitudes of employees, which enables them to work in a decent working environment. Organizations that provide green training, manage and reward employees for their green performance, and also involve employees in problem-solving and decision-making processes are expected to have employees with green competencies (e.g., green attitudes and green skills). These employees can create a decent working environment and show sustainable behavior while using organizational resources.

*5.3. Limitation and Future Research*

This study is limited to the context of the hi-tech manufacturing industry and does not include other industries that do not fall into this category of hi-tech firms. It would be challenging to generalize the results to other sectors, private or public organizations, even though the textile companies are one of the main industries in Pakistan. Therefore, future researchers need to empirically test the models in other developing and developed countries and with a different business environment. Additional longitudinal studies are also encouraged to understand sustainable consumption and production behavior over time. Future studies must use different sets of GHRM practices and SDGs to explore the association and use of other moderators, such as green climate and green culture. It is suggested that future studies should try to explore the individual GHRM practice's roles in the development of greener employee attitudes to elevate the sustainable consumption behavior. It is further suggested that other SDGs may be added to the above framework to explore the key role played by GHRM in achieving SDGs.

**Author Contributions:** Writing—review & editing, Validation, J.L.; Methodology, X.G.; Conceptualization, Y.C.; Writing—original draft, N.M.; Supervision, J.C.; and Writing—review & editing, L.W. All authors have read and agreed to the published version of the manuscript.

**Funding:** This research received no external funding.

**Institutional Review Board Statement:** Not applicable.

**Informed Consent Statement:** Not applicable.

**Data Availability Statement:** Not applicable.

**Conflicts of Interest:** The authors declare no conflict of interest.

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
