# Peer review of "Catalytic Effect of Green Human Resource Practices on Sustainable Development Goals: Can Individual Values Moderate an Empirical Validation in a Developing Economy?"

_sustainability, doi:10.3390/su142114502_

Round 1

Reviewer 1 Report

Some comments and suggestions:

11.  Authors should deal with the sustainability citation style.

22. The introduction should be updated with more recent studies at least 2020, 2021, 2022.

33. When the authors explain about the theory of planned behavior, the authors should refer to the Ajzen (1985, 1991, 2020), especially on introduction.

Ajzen, I. (1985). From Intentions to Actions: A Theory of Planned Behavior. Action Control, 11–39. doi:10.1007/978-3-642-69746-3_2

Ajzen, I. (1991). The theory of planned behavior. Organizational Behavior and Human Decision Processes, 50(2), 179–211. doi:10.1016/0749-5978(91)90020-t

Ajzen, I. (2020). The theory of planned behavior: Frequently asked questions. Human Behavior and Emerging Technologies. 2(4), 314–324. doi:10.1002/hbe2.195

4.     “The present study provides a unique combination of relationships between GHRM and SDGs, which is a missing in previous literature to date. This study contributes to the literature of GHRM and SDGs by providing a more holistic model based on the theory of planed behavior (TPB) for understanding the relationship between GHRM practices and sustainable development goals (SDGs) through Green attitude and behavior of employees in the presence of Green individual values as a moderator. Hence, the current study not only theoretically contributes but also empirically validates the GHRM -sustainability link from a different theoretical lens of (TPB).”

 The above statements do not specify which gap in the literature leads to the definition of the overall goal.

What do you mean “a more holistic model based on the theory of planed behavior (TPB)?”

 5.     The first subtitle in the theory and hypothesis should be: 2.1 Theory of Planned Behavior

6.     It is necessary to add reference support in developing hypotheses, especially hypothesis 2.

7.     Research framework in Figure 1 is too poor. Please revise the framework.

8.     Research framework should be revised with put each hypothesis in the path (arrows).

9.     Research framework should be indicated the mediating and moderating sign through their arrows (e.g., using dash arrows               )

10.  What do you mean? H7: Employee Skills and Sustainable Consumption and Production Behavior (SCPB)

 11.  Since only 197 useable completed surveys were accepted for further research, therefore authors should use bootstrapping technique to investigate direct and indirect effects (see: Hayes, 2013).

12.  Do authors used a 5-point Likert or 7-point scale to assess indicators?

13.  The authors should provide descriptive statistics of the enterprises.

14.  Do authors used reflective or formative measurement in their structural model?

15.  The authors should provide the goodness-of-fit (GoF) index of the structural model (GoF = )

16.  The authors should provide the mediating proportion.

17.  The authors should provide Sobel tests to support the indirect (mediated) effect results.

18.  In the moderating analysis, authors should provide F2 results of each figure.

19.  I am wondering, can your research be generalized?

Author Response

Additional Comments:

 Rev.1.Q.5. The authors should provide the goodness-of-fit (GoF) index of the structural model (GoF = √AVE x R2)

Author Comment: the reason The goodness of fit (GoF)was not used to the following reasons, GOF has been developed as an overall measure of model fit for PLS-SEM. However, as the GoF cannot reliably distinguish valid from invalid models and since its applicability is limited to certain model setups, researchers should avoid its use as a goodness of fit measure. The GoF may be useful for a PLS multigroup analysis (PLS-MGA) . 

  • Hair, J. F., Hult, G. T. M., Ringle, C. M., and Sarstedt, M. (2017). [A Primer on Partial Least Squares Structural Equation Modeling (PLS-SEM)(http://www.pls-sem.com), 2^nd^ Ed., Sage: Thousand Oaks.

  • Henseler, J., and Sarstedt, M. (2013). Goodness-of-Fit Indices for Partial Least Squares Path ModelingComputational Statistics, 28(2): 565-580.

  • Tenenhaus, M., Amato, S., and Esposito Vinzi, V. (2004). A Global Goodness-of-Fit Index for PLS Structural Equation Modeling, Proceedings of the XLII SIS Scientific Meeting. Padova: CLEUP, 739-742.

Rev.1.Q.17. The authors should provide Sobel tests to support the indirect (mediated) effect results.

Author Comment:  Sobel Test has distributional Assumption while bootstrapping is Assumption free. Using the Sobel test with Smart PLS is discouraged. is Hair, J. F., Hult, G. T. M., Ringle, C. M., and Sarstedt, M. 2014. A Primer on Partial Least Squares Structural Equation Modeling (PLS-SEM). Thousand Oaks, CA: Sage.

Rev.1.Q.19 I am wondering, can your research be generalized?

Author Comment:  I am thankful for your valuable comment. Yes, the research could be generalized in other settings and will provide us insight into GHRM practices that hinder or facilitate the implementation of SDGs.

Reviewer 2 Report

1. provide clear research framework, along with moderator and mediators clearly showing the direction of arrows. 

2. Some of citations in reference list are incomplete, add their respective volume and issue numbers and page numbers. 

3. Add HTMT ratios for discriminant validity. 

4. it is suggested to remove those loadings which are less than 0.7 then report their CR, AVE and Cronbach alpha. 

5. moderations graph are ok, but if graph produced by PLS are added, it will be better. 

6. in framework it is shown that GHRM is taken as higher order while discriminant analysis is conducted on lower order (facets wise). it must be consistent i.e. on higher order. 

Author Response

  1. provide a clear research framework, along with moderator and mediators clearly showing the direction of arrows.

Author Comment:  Thank you for your comment . The research framework is reproduced in light of the above comment. Please see the attached file.

2. Some of citations in reference list are incomplete, add their respective volume and issue numbers and page numbers. 

Author Comment: Thankyou. We have added the requisite information

3. Add HTMT ratios for discriminant validity. 

Author Comment: Thankyou. We have added the Table.

4. it is suggested to remove those loadings which are less than 0.7 then report their CR, AVE and Cronbach alpha. 

Author Comment: Thank you for your valuable comment. Some items if retained, were to make sure that at least 3 items are attached to each construct as per Smart PLs. Secondly, the item was important in the context of the study and hence was not dropped. 

5. moderation graph are ok, but if the graph produced by PLS are added, it will be better. 

Author Comment: Thank you for your suggestion. 

Reviewer 3 Report

I enjoyed reviewing this manuscript and it can stand the test of time. However, there are several areas of concern that should be addressed. 

1. The research gap(s) are not clearly highlighted.

2. Why are these research outcomes important that it calls for empirical research?

3. Except for the first hypothesis, all other hypotheses are inadequately developed. A logical and clear derivation is required.

4. The methodology should be detailed enough to be replicated by other researchers.

5. The plausible reasons for the hypotheses not being supported should be discussed. The discussions should be elaborated in light of existing literature.

6. Theoretical and practical implications are too narrow.

7. Limitations and areas of further research should be adequately explored.

Author Response

1. The research gap(s) are not clearly highlighted.

Author Comment:  Thank you for your valuable comment. We have revised the write-up in light of the above comment. Please see the attached file.

2. Why are these research outcomes important that it calls for empirical research?

Author Comment:  We have addressed the issue in the introduction part. 

3. Except for the first hypothesis, all other hypotheses are inadequately developed. A logical and clear derivation is required.

Author Comment:  Thank you for highlighting it. We have addressed the issue in light of available literature and theory. Please see the attached file.

4. The methodology should be detailed enough to be replicated by other researchers.

Author Comment: The methodology portion is improved and was required to address the issue. Please see the attached file.

5. The plausible reasons for the hypotheses not being supported should be discussed. The discussions should be elaborated in light of existing literature.

Author Comment: The discussion is improved in light of available literature.

Limitations and areas of further research should be adequately explored.

Author Comment:  Limitation and future avenues are addressed. Thankyou for your comment.

Round 2

Reviewer 3 Report

I appreciate the author(s) for addressing all the concerns raised. Best wishes.